# Nano-Scale Residual Stress Profiling in Thin Multilayer Films with Non-Equibiaxial Stress State

**DOI:** 10.3390/nano10050853

**Published:** 2020-04-28

**Authors:** Marco Sebastiani, Edoardo Rossi, Muhammad Zeeshan Mughal, Alessandro Benedetto, Paul Jacquet, Enrico Salvati, Alexander M. Korsunsky

**Affiliations:** 1Engineering Department, Università degli studi Roma Tre, via della Vasca Navale 79, 00146 Rome, Italy; 2School of Engineering & Innovation, The Open University, Walton Hall, Milton Keynes MK7 6AA, UK; 3Saint-Gobain Research Paris, 39 Quai Lucien Lefranc, 93303 Aubervilliers Cedex, France; 4Polytechnic Department of Engineering and Architecture (DPIA), University of Udine, Via delle Scienze 208, 33100 Udine, Italy; 5Multi-Beam Laboratory for Engineering Microscopy (MBLEM), Department of Engineering Science, Parks Road, Oxford OX1 3PJ, UK

**Keywords:** FIB-DIC, ring-core, low emissivity coatings, residual stress, profiling, non-equibiaxial stress, adhesion

## Abstract

Silver-based low-emissivity (low-E) coatings are applied on architectural glazing to cost-effectively reduce heat losses, as they generally consist of dielectric/Ag/dielectric multilayer stacks, where the thin Ag layer reflects long- wavelength infrared (IR), while the dielectric layers both protect the Ag and act as an anti-reflective barrier. The architecture of the multilayer stack influences its mechanical properties and it is strongly dependent on the residual stress distribution in the stack. Residual stress evaluation by combining focused ion beam (FIB) milling and digital image correlation (DIC), using the micro-ring core configuration (FIB-DIC), offers micron-scale lateral resolution and provides information about the residual stress variation with depth, i.e., it allows depth profiling for both equibiaxial and non-equibiaxial stress distributions and hence can be effectively used to characterize low-E coatings. In this work, we propose an innovative approach to improve the depth resolution and surface sensitivity for residual stress depth profiling in the case of ultra-thin as-deposited and post-deposition annealed Si_3_N_4_/Ag/ZnO low-E coatings, by considering different fractions of area for DIC strain analysis and accordingly developing a unique influence function to maintain the sensitivity of the technique at is maximum during the calculation. Residual stress measurements performed using this novel FIB-DIC approach revealed that the individual Si_3_N_4_/ZnO layers in the multilayer stack are under different amounts of compressive stresses. The magnitude and orientation of these stresses changes significantly after heat treatment and provides a clear explanation for the observed differences in terms of scratch critical load. The results show that the proposed FIB-DIC combined-areas approach is a unique method for accurately probing non-equibiaxial residual stresses with nano-scale resolution in thin films, including multilayers.

## 1. Introduction

Glass is widely used in many applications such as optoelectronics, construction and decoration, medicine, automotive, aerospace, and solar energy harvesting [1,2,3,4,5,6]. Glass used in building and vehicle windows is often coated with low-emissivity multilayer thin films. These coatings offer high transmission and low emissivity, hence allowing more sunlight to pass through while blocking the infrared (IR) radiation to minimize undesirable interior heating. These low-emissivity (low-E) coatings are often formed by depositing a reflective layer (e.g., silver) onto the glass surface. To improve its electrical/optical properties, strongly depending upon its microstructure, and provide both adhesion and protection, several other layers are formed below and on top of the reflective layer, as shown in Figure 1. These layers are mostly dielectric hard materials such as silicon nitride (Si_3_N_4_), which is commonly used because of its transparency over a wide spectral range from the near ultraviolet (UV) to the infrared (IR) region, tin oxide (SnO_2_), and zinc oxide (ZnO), the latter mainly enhancing reflective layer adhesion and promoting microstructural characteristics that reduce its resistivity. These dielectric layers not only act as an optical filler and function as anti-reflective barriers to improve the optical properties, but also protect the stack from both the substrate and the environment. Both reflective and dielectric layers are normally produced using the physical vapor deposition (PVD) technique and the overall thickness of the stack is normally <300 nm. Some typical configurations and properties for these layers are widely available in the literature [7,8,9].

The mechanical properties of these coatings are extremely important as they can have a detrimental effect on the overall performance of the low-E coatings [10,11,12,13]. These properties not only define the overall adhesion of the coating to the substrate, but also determine the effect of contact loads on the coating surface and the scratch resistance. Heat treatment procedures, such as annealing and tempering, are the most used finishing processes that can enhance the resistance to external loads and in turn improve the overall performance of the coatings. Heat treatment such as tempering is normally employed after the coating process and involves heating the glass to a high temperature in the range of the glass transition (i.e., for soda-lime substrates typically 520÷630 °C) in a convection furnace, with set points either way not exceeding 700 °C to avoid coating damage, and immediately cooling from the edges and the outer surface to achieve rigid surface/edges while the core is allowed to cool down much slower as compared to the edges. This can develop compressive stresses at the surface of the glass/coating, which in turn protect the tempered glass surface under normal (compressive) deformations. 

The quantification of the residual stress is important for the understanding of the overall mechanical performance of these coatings [14]. Mostly, the average residual stresses are measured in amorphous coatings using the wafer curvature method, which is an indirect method to determine the average stress values. This method only provides an average value of the stress, and no information is obtained regarding the residual stress variation within the coating from the surface to the interface with the substrate. The detailed knowledge of the residual stress profile provides extremely useful insights into the overall coating behavior. However, no methods are available in the literature to accurately predict the residual stress profile in amorphous multilayer films on glass substrates, with an overall thickness below 300 nm and an individual layer thickness of the order of 10 nm. In addition to this, the residual stress in coated glass components can also be strongly non-equibiaxial, because of the complex geometry and large dimensions of glass substrates combined with the effects of post-deposition treatments. 

In summary, a significant lack of knowledge is still present in the literature, which is related to the ability to measure non-equibiaxial residual stress depth profiles in very thin amorphous multilayer films over glass substrates.

Incremental focused ion beam (FIB) milling combined with high-resolution scanning electron microscopy (SEM) and digital image correlation (DIC) has evolved as a new extremely useful technique in the past decade to understand the residual stress distribution at micro- to nano-scales [15] and has wide range applications in various disciplines of materials science [16,17,18]. The FIB-DIC method allows for the measurement of the relaxation strain distribution ***e(h)*** at each milling depth **h**, which can be converted into a residual stress depth profile by using suitable inverse calculation procedures and knowing the distribution of the elastic constants of the probed volume of material [19].

In recent papers [20,21], the authors have introduced an eigenstrain-based procedure for the measurement of residual stress depth profiles in coatings with a depth resolution of about 50 nm. The method assumes that the eigenstrain in the body remains constant during the incremental FIB milling process. The “eigenstrain” identifies the amount of inelastic deformation present in the probed material, which is generated during its manufacturing or post-manufacturing processes. Eigenstrain can be defined as the invariant source of the residual elastic strain field necessary to keep balance of forces and moments within the body, according to the equilibrium conditions. This assumption is valid provided that both the local curvature and the thickness of the probed materials are small enough. 

Using this assumption, the strain relief at a generic normalized milling depth *h*/*D*—being *h* the depth reached after each FIB material removal increment and *D* the ring-core pillar diameter—can be expressed as follows:(1)$$e\left(h/D\right)=\int_{o}^{\frac{h}{D}}F\left(\frac{z}{D}\right)\varepsilon^{*}\left(z\right)\;d\left(\frac{z}{D}\right)$$

Here, ***ε*(z)*** is the unknown distribution of the eigenstrain at the calculation depth ***z***; therefore, ***z/D*** is the normalized depth at which eigenstrain is being determined, and ***F(z/D)*** is the calibration influence (or sensitivity) function, which is usually calculated by Finite Element Analysis.

For the case of the non-equibiaxial residual stress state, separate influence functions for the hydrostatic and deviatoric parts should be calculated, as described in a previous paper by the authors [21].

A careful analysis of the strain distribution over the pillar surface during relaxation shows that the calculated influence function can be significantly different depending on the fraction of central pillar area that is selected for DIC analysis. In more detail, the *area%* for DIC is determined by selecting a certain percentage of the pillar’s radius (from here on called *radius%*, see Figure 2a,b). This corresponds to different sensitivity functions ***F(z/D)*** that are reported in Figure 2c,d, which show that a maximum exists for each of the sensitivity functions at the different values of the *radius%* selected for calculation. In order to avoid inaccuracies during stress calculation, a value of 80% was suggested to guarantee a sufficiently high strain sensitivity in the depth range of 0.015 < ***h/D*** < 0.2 for both the hydrostatic and deviatoric stress components.

Such functions show clearly that the sensitivity of the method to an eigenstrain depth variation decreases remarkably for ***h*/*D*** values higher than 0.2 for all considered values of the *radius%*. Therefore, the range of 0.015 < ***h/D*** < 0.2 for effective depth profiling has been suggested for optimal residual stress depth profiling. In this way, a depth resolution of 50 nm was demonstrated for pillar diameters in the range of 1–15 µm [20].

Equation (1) represents the basic formulation for the inverse calculation of the residual stress depth profile, according to the procedure described in a previous paper by the authors [20].

In case of coating/substrate systems, the maximum milling depth is usually set to be equal to film thickness. On this basis, the main idea for expanding the maximum milling depth for depth profiling was to combine results from pillars with multiple diameters (multi-pillar approach) in order to be able to maintain a sufficiently high stress sensitivity even for large milling depths [20,22]. However, the multi-pillar approach cannot be used in case of very thin multilayer films (< 300 nm) with an individual layer thickness in the range of 10–40 nm, since the pillar diameter should not be smaller than 1 µm (in order to reduce possible artefacts and damage from FIB milling [23]). Therefore, there is still a lack of both depth resolution and surface strain sensitivity for the analysis of residual stress gradients in very thin films. Additionally, the calculated influence functions are only usable for homogeneous coatings, and an extension for the case of multilayer stacks is needed.

In the present study, we propose a substantial improvement of the micro ring-core method for further enhancement of the depth resolution for residual stress depth profiling and extending the maximum depth to ***h*/*D*** = 0.25 in multilayer amorphous/crystalline films on a glass substrate. The main idea is to develop a unique influence function by considering different percentages of area (i.e. different *radius%* of the pillar) for the DIC analysis, in order to keep strain sensitivity always at its maximum during the calculation. In this way, we demonstrate nano-scale non-equibiaxial residual stress profiling in ultra-thin films (thickness < 300 nm) with an individual layer thickness of the order of 10–50 nm.

Finally, the method is applied to an Si_3_N_4_/Ag/ZnO multilayer stack on a glass substrate, where the influence of residual stress depth gradient on film adhesion (scratch critical load) is clearly identified (for the same average stress and hardness of the film). The results show that the nanoscale residual stress depth profile can be the main design parameter to be controlled for the optimization of adhesion in multilayer low-emissivity thin films on glass substrates.

## 2. Materials and Methods 

### 2.1. A New Approach for Improving the Sensitivity and Resolution of Residual Stress Depth Profiling in Very Thin Films 

The accuracy, resolution, and sensitivity of residual stress depth profiling by the FIB-DIC micro ring-core configuration depends on the proper choice of pillar diameters in relation to the film thickness and the adoption of the optimal area percentage (corresponding to an optimal *radius%*) of the central pillar during digital image correlation, as discussed in previous papers by the authors [20,21]. 

In this paper, we propose a new idea for improving the strain sensitivity, resolution, and maximum achievable depth during residual stress depth profiling in very thin multilayer films. The idea comes from the observation that the strain sensitivity functions reported in Figure 2c,d can differ significantly depending on the *area%* (or *radius%*) used for strain calculation (Figure 2b). 

In fact, the curves show that the optimal choice of the *radius%* actually depends on the considered range of the relative milling depth ***h*/*D***. Higher values (>80%) of *radius%* should be used for ***h*/*D*** < 0.05, while the adoption of lower values (<70%) should be recommended for h/D>0.1. This observation can be converted in the simple area selection scheme reported in Figure 3, where the optimal *radius%* to be used is plotted as a function of ***h*/*D*** ratio (for both the deviatoric and hydrostatic calibration functions) precisely basing on the relative milling depth ranges for which the corresponding strain sensitivity functions values are higher than the others.

It is important to note that *area%* (or *radius%*) close to 100% is not experimentally achievable because of the FIB-induced damage at the pillar’s edge and/or FIB re-deposition occurring during the milling process. Therefore, the use of the multiple *area%* is expected to improve, at the same time, (i) the surface sensitivity, (ii) the depth resolution, and (iii) the maximum usable depth for residual stress depth profiling by the FIB-DIC micro ring-core method. In this way, the strain sensitivity can be improved up to the maximum at the shallowest depths and up to 30% for the maximum value at ***h***/***D*** = 0.2. By doing so, a sufficiently high value of strain sensitivity can be achieved even at ***h*/*D*** = 0.01 (by using *radius%* = 85% at the shallowest depths), and the maximum depth can be increased to ***h*/*D*** = 0.25 (by using *radius%* = 60% at the deepest milling depths).

In the case of multilayer films, discontinuities in material properties, notably the elastic modulus, at each interface from one layer to another, occur at specific values of the h/D parameter, depending upon the coating architecture. The transitions from one *radius%* value to the next one, corresponding for the *optimal radius%* in Figure 3 to the vertical lines at definite values of h/D, can be set to occur at the values of the relative milling depth at which the discontinuities for the Young’s modulus take place, moderately offsetting from the *optimal radius%* curves. In this way, mathematical calculation singularities of residual stresses that could exist due to any change in the *optimal radius%* value—and, consequently, in the definition of the corresponding strain sensitivity function values—occurring at h/D ratios falling within a layer are avoided and engrossed in the discontinuity of material property, still maintaining improved strain sensitivity capabilities for the technique.

### 2.2. Coating Deposition and Heat Treatment 

Deposition of the coatings has been performed using an industrial coater on large (nominally flat) glass substrates with PLF dimension (Plateau Largeur de Fabrication, corresponding to 6 m × 3.21 m) and a thickness of 6 mm. Deposition was performed by magnetron sputtering physical vapour deposition (MS-PVD). The vacuum inside the chamber was below 30 µbar. The peculiar shape (large area) and process (continuous PVD deposition) resulted in strongly non-equibiaxial stress fields in the films after deposition.

The coatings deposited for the present study are simplified (but representative) models of actual solar control low-E stacks, which normally consist of a multitude of thin films (see Figure 4). The stack has an overall thickness of 270 nm and is characterized by the following structure, starting from the substrate: glass substrate (6 mm)/silicon nitride (30 nm)/zinc oxide (10 nm)/silver (15 nm)/zinc oxide (10 nm)/silicon nitride (60 nm)/zinc oxide (10 nm)/silver (10 nm)/zinc oxide (10 nm)/silicon nitride (60 nm)/zinc oxide (10 nm)/silver (10 nm)/zinc oxide (10 nm)/silicon nitride (30 nm).

The stack was deposited by passing under successive sputtering targets (without exiting the chamber). Metallic targets were used as the sputtering target, while argon was used as the plasma-forming gas. For the deposition of silicon nitride layers, reactive sputtering was used with the injection of nitrogen in the plasma. A similar thin film architecture was also deposited (using the same deposition parameters and layer thickness sequence) on a thin glass substrate, in order to be able to measure the average residual stress in the film by means of the curvature (Stoney’s equation) method. Since this approach does not require any information on the elastic properties of the film, the measurement gives information on the average residual stress throughout the entire film thickness.

The coated glasses were tempered in an industrial furnace. Glass plates with dimensions of 400 mm × 400 mm were placed on conveyor rollers and moved inside the furnace while being heated to a temperature above the glass transition temperature (approximately 540 °C) for 12 minutes. Then, the samples were taken out of the furnace and quickly cooled down to room temperature using air fans. The quality of the tempering was checked according to the breakage pattern: no fragment had a lateral dimension exceeding 10 mm.

### 2.3. Nanoindentation and Nanoscratch Testing 

Nanoindentation and nanoscratch testing were performed on Nano indenter® G200 (KLA Corporation, Milpitas, CA, USA) using a Berkovich tip. A scratch of 100 µm length was made on all samples with a maximum scratch load of 20 mN while maintaining a velocity of 20 µm/sec. At least 5 scratches were performed on each sample, and the results are reported as average values corresponding to the critical loads of adhesive failure of the films, which were identified by the analysis of the displacement versus scratching load graphs, as well as by SEM and 3D optical profilometry observations of the tracks. The nanoindentation experiments were performed using the Continuous Stiffness Measurement (CSM) mode, in order to acquire the information on the depth variation of hardness and elastic modulus for the multilayer systems under investigation. The machine frame stiffness and tip area function were calibrated by performing measurements on a certified fused quartz reference sample, according to ISO 14577 standard. Given the complex architecture of the film, it was not possible to have a statistically robust evaluation of the elastic modulus values of the individual layers of the film. Therefore, the results obtained by nanoindentation provided an estimation of the average composite hardness and modulus for the entire film, which were calculated at a representative indentation depth of 30 nm.

### 2.4. Residual Stress Depth Profiling 

FIB cross-sectioning and residual stress profiling were carried out in a Helios Nanolab 600 dual beam FIB/SEM (Thermo Fisher Scientific, Waltham, MA, USA). The ring-core procedure [15,19,21] was performed using a current of 28 pA at 30 kV. Incremental material removal (i.e., step-by-step milling) of the ring-core geometry was carried out using a semi-automated procedure and an annular trench of 1.5 µm inner diameter. Each milling step reached a depth of the order of 7 nm, which was achieved by a low ion milling current and by careful calibration of the milling depth versus ion dose for the specific material under investigation via measurements on cross-sections performed over the milled geometries. Using this approach, the milling rate was kept constant during the entire milling process, which ensures that the partial depth reached after each incremental material removal is uniform and has constant values. In addition, the final depth reached, which is about the coating thickness to evaluate the residual stress depth profile from the surface to the interface with the substrate, is derived from the total number of steps. Cross-checking via cross-sectional depth measurements on SEM high-resolution images of the milled geometries lateral surfaces have been also performed. In this way, the film thickness corresponds to an overall relative milling depth of ***h*/*D*** = 0.18. Five high-resolution SEM micrographs were acquired prior to testing (reference surface images, namely *step #0* in Figure 5) and after each of the material removal increments (step), maintaining the same contrast as the reference images, using an acceleration voltage of 5 kV and a beam current of 0.34 nA, integrating for each image 128 frames at 50 ns dwell time each.

The micrographs were acquired in three different orientations (0°, 45°, and 90°) using the scan rotation option in the SEM microscope (Figure 5). In this way, the strains are always measured along the fast-scan direction, which is less affected by scanning and acquisition artefacts [24]. Both electron and ion beam drifts were automatically corrected during the test. A MATLAB™-based DIC routine [25] was used to obtain the relaxation strain profiles from the micrographs at each milling step, with respect to the original image of material surface prior to milling. An area of correlation markers—the center point of a subset, that is the part of the image which will be correlated—has been defined, corresponding to the pillar’s central area (with the proper *area%* reduction) with a spacing of 8 pixels in the two orthogonal in-plane directions and a subset size of 50×50 pixels. In this way, the non-equibiaxial relaxation strain was calculated using three components of strain in the 0°, 45°, and 90° directions. In the general case, the three components of the incremental strain relief were used to calculate the two in-plane principal stress depth profiles and their orientation with the respect to the reference coordinate system, according to the procedure as reported in a previous paper by Salvati et al. [21].

To cope with the experimental artefacts described in Section 2.1, a possible range of 60–85% for *radius%* is considered in this work. Only four intervals of influence functions with different *radius%* are considered along the film thicknesses: 0–40 nm (85%), 40–130 nm (80%), 130–220 nm (75%), and 220–250 nm (60%). As proposed in Section 2.1, the discontinuities of *radius%* are set to be exactly in correspondence to some of the elastic modulus discontinuities. The range is valid for both hydrostatic and deviatoric influence functions.

As discussed in the previous section, the results from nanoindentation did not allow for a statistically robust estimation of the elastic modulus of individual layers. Although this is possible in principle, by applying available models on the CSM elastic modulus profiles, it becomes a very challenging task as soon as individual layer thickness is of the order of 10 nm. For this reason, in order to calculate residual stress profiles, suitable elastic modulus and Poisson’s ratio values of individual layers were taken from the literature: for both the dielectric layers materials (${\mathrm{Si}}_{3}N_{4}$ and ZnO) values of 120 GPa and 0.3 for the elastic modulus and Poisson’s ratio respectively, while for the Ag reflective layer, there were values of 80 GPa and 0.3 [10,11].

The error bars of residual stress profiles were only calculated based on the statistical error on the relaxation strains. Additional sources of error arise due to the uncertainty on the elastic modulus and the actual milling depth; assuming that the material removal rate remains constant through the milling process represents an approximation.

In the following diagram (Figure 6), the entire process is described (from ring-core selection to final stress calculation). The choice of the optimal core diameter should be done with reference to the actual film thickness; in addition, the calculation steps and the different *radius%* to be used are selected in relation to the multilayer architecture of the film.

### 2.5. Validation by Residual Stress Curvature Measurement

For an independent validation of residual stress measurements, the curvature-based method was used to estimate the overall residual stress in the film. To obtain information about the stress depth profile, the deposition was also performed on thin glass substrates (dimensions 9 mm × 13 mm × 0.4 mm from PGO GmbH, Iserlohn, DE) to allow better sensitivity and enable the direct measurement of sample curvature.

The topography of the surface has been measured before the deposition process using a NewView™ 9000 interferometric microscope (Zygo Corporation, Middlefield, CT, USA) at a magnification of 1x with a zoom factor of 0.5x. Samples were placed on a customized stage having three contact points constituted of small spheres of 1 mm diameter to obtain reproducible positioning.

Then, the topography has been measured after the deposition of each layer, which composes a stack sequence of the coating with respect to the thin glass substrate and single layer data combined to obtain the overall average stress, as detailed further on. Finite differences are used to combine the stacks calculated average stresses and obtain the depth profile for the sequence.

In order to obtain the surface change due to the stressed film, measurements after a deposition were subtracted to the topography evaluated before.

Data was fitted by least squares minimization with the expression:(2)$$z=z_{0}+A_{1}x+A_{2}x^{2}+B_{1}y+B_{2}y^{2}+Cxy$$
which allows constructing the second fundamental form of the surface as:(3)$$II=\left(\begin{array}{cc} 2A_{2} & C\\ C & 2B_{2} \end{array}\right).$$

Then, the principal curvatures of the surface are extracted as the eigenvalues of the second fundamental form and the principal curvature directions are extracted as the eigenvectors of that form.

Stoney’s equation is used to calculate the average stress in the thin film from the differential curvature data:(4)$${\bar{\sigma }}_{f}=\frac{1}{6}{\left(\frac{E}{1-\nu }\right)}_{s}\frac{t_{s}^{2}}{t_{f}}k$$
where ${\left(E/\left(1-\nu \right)\right)}_{s}$ is the biaxial modulus of the substrate, $t_{S}$ and $t_{f}$ are the thickness of the substrate and of the film, and *k* is the curvature of the substrate measured as described.

This equation cannot be used to extract the stress in the case of a non-equibiaxial stress state. Zhao et al. [26] extended the Stoney’s equation in case of anisotropic stress as:(5)$$\bar{\sigma_{f,1}}=\frac{1}{6}{\left(\frac{E}{\left(1-\nu \right)\left(1+\nu \right)}\right)}_{s}\frac{t_{s}^{2}}{t_{f}}\left(k_{1}+\nu_{s}k_{2}\right)$$
(6)$$\bar{\sigma_{f,2}}=\frac{1}{6}{\left(\frac{E}{\left(1-\nu \right)\left(1+\nu \right)}\right)}_{s}\frac{t_{s}^{2}}{t_{f}}\left(k_{2}+\nu_{s}k_{1}\right)$$

$\bar{\sigma_{f,1}}$ and $\bar{\sigma_{f,2}}$ are the average principal stresses in the film, while $k_{1}$ and $k_{2}$ are the principal curvatures of the substrate. These equations are valid for a substrate behaving as elastically isotropic. 

With the assumption that the residual stress state of the inner layers does not change after the deposition of additional layers, the average stress in a stack is given by:(7)$${\bar{\sigma }}_{f}=\frac{\sum_{i}\sigma_{i}t_{i}}{\sum_{i}t_{i}}.$$

This aspect can be a limitation of the curvature method for measuring the residual stress gradients in coatings, since the stress state may change during the growth of additional layers and the final cooling.

Here, $t_{i}$ and $\sigma_{i}$ are respectively the thickness and the stress of each individual layer. In a coating composed by a stack of layers 1, 2, …, M and a second stack of layers 1, 2, …., M, … N, the average stress in the stack from M + 1 to N is calculated as:(8)$$\bar{\sigma_{M+1\to N}}=\bar{\sigma_{N}}+\bar{\sigma_{N}}\frac{t_{M}}{t_{M+1\to N}}-\bar{\sigma_{M}}\frac{t_{M}}{t_{M+1\to N}}$$
where $t_{M}$ is the total thickness of the stack 1, 2, …, M and $t_{M+1\to N}$ is the total thickness of the stack of the layers M + 1 … N, $\bar{\sigma_{N}}$ is the average stress in the stack 1, 2, …., M, … N, and $\bar{\sigma_{M}}$ is the average stress in the stack 1, 2, …, M. The analysis is performed independently in the two principal directions with the assumption that their orientations are not modified after each deposition step as they correspond to the directions perpendicular and parallel to the sweeping direction under the sputtering targets.

## 3. Results

The results of nanoindentation and scratch testing for samples A (after heat treatment) and B (before heat treatment) are reported in Figure 7 and Table 1. It is apparent from the depth profiles presented in Figure 7a,b that in terms of the elastic modulus, no significant differences were observed between the two samples (Figure 7a). On the contrary, a statistically significant reduction in hardness is seen in the depth profile plot after heat treatment (Figure 7b). Film adhesion evaluated from nanoscratch results (Figure 7c) shows a reduction after heat treatment, as seen for the values of ***L_c2_*** (first adhesive failure) and ***L_c3_*** (full coating delamination) critical loads. In particular, a very significant difference in terms of ***L_c2_*** critical load is observed. The two failure events are visible in both cases from the SEM images reported in Figure 7c,d.

Figure 8a shows the results of FIB-DIC analysis in the form of relief strain values in three directions (designated 0°, 45°, and 90°) in sample B as a function of the milling depth. The difference between the three directions is apparent from the strain relief curves, with the 0˚ curve appearing particularly different. The difference in the strain magnitude increases toward the interface with the substrate (250 nm) where the significantly non-equibiaxial stress state prevails.

In sample A after heat treatment, relief strains appear to be more similar in all directions, thus suggesting an equibiaxial stress state. However, some differences in the proximity of the film/substrate interface are also observed in this case (Figure 8b).

Figure 8c,d presents the reconstructed residual stress profiles in samples B (before) and A (after heat treatment). The presence of non-equibiaxial stresses is apparent at the surface of sample B before heat treatment. The stress state remains compressive for the entire film thickness and shows high compressive stress states both at the surface and the interface. These observations are consistent with the expected growth mechanisms and structure/stress correlation for sputtered films. In fact, a strong interfacial compressive stress can be expected at the early stage of film growth, as reported in previous papers [27,28]. Figure 8c also reports the comparison between FIB residual stress profile and the equivalent stresses calculated from the curvature measurement at four different steps of coating deposition. The agreement is notably good in terms of the average stress values. The curvature method captured the non-equibiaxial nature of the stress state only to a small extent, whereas the FIB-DIC results highlight this aspect clearly.

The high surface residual stress can be related to inhomogeneous cooling that may have occurred during very large area deposition adopted in this study. Our previous studies showed that (i) critical load during nanoscratch depends strongly on the contact radius of the counterpart during wear and that (ii) depending on the radius of the counterpart and the contact radius, the depth at which the compressive stress should be maximum in order to have improved scratch resistance will change. Based on those results, it is expected that a coating with tensile interfacial residual stress (Figure 6d) will show a lower scratch critical load.

It is clear from Figure 8d that a remarkable change of the residual stress depth profile occurs after the heat treatment. Stress becomes almost equibiaxial for most of the film thickness following heat treatment due to microstructural rearrangement and homogenization in the film. The stress state changes are particularly notable at the top surface and at the coating/substrate interface, where tensile residual stress appears. This could be related to the glass softening during heat treatment that gives rise to compressive stress in the substrate and tensile stress in the coating close to the interface.

High tensile interfacial stresses in the presence of strong gradients are likely to cause a reduction in thin film adhesion. Nanoscratch results reveal that sample A shows significantly lower critical loads ***L_c2_*** and ***L_c3_*** values. Specifically, a remarkable decay of ***L_c2_*** is observed that corresponds to surface delamination. This can be directly correlated to the measured tensile surface stress in sample A after heat treatment (Figure 8d).

## 4. Discussion

The main purposes of the present work were to demonstrate that (1) improved depth resolution can be achieved by the incremental micro-ring-core FIB-DIC method, and (2) such method can be relevant for improving the understanding of the nanoscale mechanical behavior of multilayer thin films. A remarkable change of the stress profile after the heat treatment was observed for Si_3_N_4_/Ag/ZnO multilayer films, and a clear correlation with nanoscratch critical loads results was identified.

FIB-DIC residual stress profiles were in good agreement with the independent results obtained from curvature measurements at different steps of the coating deposition. Results from experiments show that accurate relaxation strain profiles along multiple directions can be achieved (with individual steps of the order of 7 nm) by using the developed FIB-DIC protocol. 

Although the experimental milling step was of 7 nm, the actual resolution in terms of residual stress profiling is also related to other calculation and fitting parameters, e.g., the approach used for strain fitting that involved a degree of smoothing of the strain data, Figure 8a. Practical resolution of the order of approximately 20 nm was achieved, as detailed below.

By using multiple *area%* (selected by a corresponding *radius%*) for relaxation strain mapping in combination with FE analysis, it has been demonstrated that strain sensitivity can be maximized by using higher values of *radius%* for the shallowest depths and lower values for the deeper points. Experimental data also suggest that the theoretical range for optimal *radius%* (100–60%, Figure 3) is not fully applicable in practice because of issues related to FIB milling damage and material redeposition, so that robust data can only be obtained in the range of 85–60%. Additionally, discontinuities in terms of influence functions used were collocated with discontinuities in the elastic modulus, as described in the previous section. The stress evaluation results capture differences between different layers of the Si_3_N_4_/Ag/ZnO nano-multilayer films. By considering all those aspects, resolution below 20 nm was achieved on an average, with improved surface sensitivity (down to ***h/D*** = 0.01) and extended maximum allowable depth (up to ***h/D*** = 0.25), in comparison with our previous paper [20].

To further describe this aspect, one may consider the example of Figure 9a–b, where the differences of (a) strain profiles and (b) stress calculations for different *radius%* values are reported. In particular, using 85% could capture the surface stress gradient for the sample after heat treatment, whilst the use of lower *radius%* values does not capture the tensile stress at the film surface. 

The first main discussion point is related to the choice of the ring-core geometry (and a single core diameter) for the specific case under investigation, which consists of multilayer thin films with thickness <300 nm. In this case, the possibility of using pillars with multiple diameters (as explored in detail in previous papers from the same authors [20]) is not feasible, given the possible artefacts coming from the FIB damage of pillars with diameter <1 µm [23]. That is the reason why we arrived at the idea of using multiple values of *radius%* (i.e. multiple pillar’s area fractions), which allows for tuning the stress sensitivity at different depths and/or in correspondence of different layers of the film. This choice lead to a clear improvement of the data quality at very shallow depths and to a very good agreement of the stress data obtained. Both the surface sensitivity and the maximum achievable depth for the method were improved in this way. A recent review paper from some of the authors [29] considered different trench shapes and concluded that circular geometry offers significant advantages for residual stress depth profiling for non-biaxial situations. In fact, linear trenches are unlikely to deliver the sensitivity sought in this study, as relief strain is highly non-uniform, and the mathematical framework to extract the depth variation of eigenstrain/residual stress is not available.

Another important point of discussion concerns the validation of obtained residual stress profiles. The average stress from curvature reported in Table 1 is in good agreement with the average stress values calculated from FIB-DIC results as the average from stress profiles reported in Figure 8c and in Table 1. The variation of stress magnitude is greater in FIB-DIC results compared to the macro-scale curvature method [19,20], since the representative volume element (RVE) is extremely different for the two cases [30]: FIB-DIC only probes a few cubic micrometres (µm^3^), whilst curvature monitoring interrogates the response of entire sample and also includes stress relaxation effects because of microdefects and localized film delamination. Additionally, it is important to remind that the curvature method is based on the strong assumption that the residual stress state of the inner layers does not change after the deposition of additional layers. This is a limitation of this method and it can be a main explanation as to why the curvature method is not able to fully capture the non-biaxiality of the stress. Additionally, the non-biaxiality could also arise during the final cooling of the large area (thick) substrates.

Further robust validation could be obtained by using nanodiffraction methods, as reported in a recent paper for a 3 µm TiN PVD coating [20]. However, this technique cannot be used in this case, since some of the layers are non-crystalline or amorphous.

This comparison suggests that for thin multilayer coatings, evaluating only the whole thickness average residual stress is not enough to elucidate the mechanisms that determine film integrity and adhesion. In fact, the loss of adhesion observed in this case is mostly due to the presence of a tensile residual stress either at the top surface or the film/substrate interface that would not be detected by average stress measurement e.g., by the curvature method.

The results presented here demonstrate that the scratch critical load of very thin films on glass can show stronger dependence on the residual stress gradient, rather than the average stress value. The minimization of the surface and interfacial residual stress can be chosen as key design variable for the optimization of adhesion and durability of ultra-thin nano-multilayer films on glass substrates.

## 5. Conclusions

This article presented an improved procedure for nano-scale residual stress evaluation in multilayer thin films with non-equibiaxial stress states. The use of focused ion beam incremental nano-scale milling experiments was combined with optimized digital image correlation (DIC) and eigenstrain analysis procedures. The main innovative idea involves the use of different fractions of the pillar’s area (i.e. *radius%*) for DIC evaluations at different milling depths to maximize the strain sensitivity and improve the residual stress estimation at deeper locations of the probed film. The results obtained for Si_3_N_4_/Ag/ZnO nano-multilayer films on glass substrate before and after heat treatment show that the method can capture the fine detail of the changes of the residual stress nano-scale gradients following thermal processing. The residual stress profiles evaluated by FIB-DIC agreed well with the results of nanoscratch testing, which confirmed that the reduction of scratch critical load can be correlated with the presence of tensile residual stresses at the film/substrate interface. The observations demonstrate that the adhesion of thin multi-layer films on glass can be significantly affected by the residual stress gradients, which should be used as a key design parameter for complex coated systems. 

## Figures and Tables

**Figure 1 nanomaterials-10-00853-f001:**
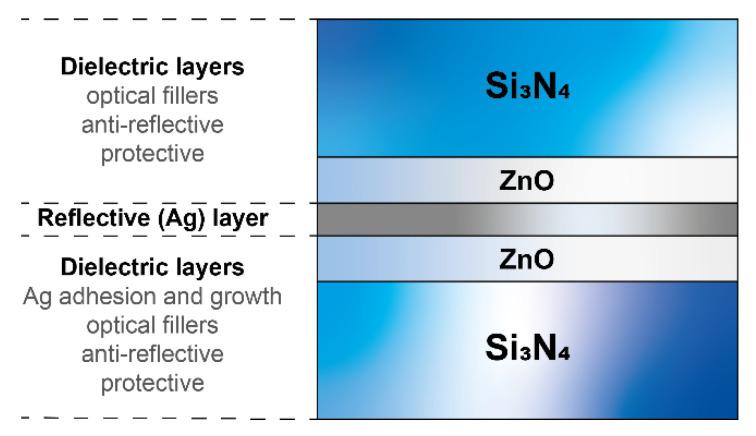
Typical silver-based low-emissivity (low-E) coating five-component stacking scheme. The thickness of the reflective layer has values of about 10 nm, while that of the dielectric layers could vary based on designed performances for the film. Coatings with multiple reflective layers are deposited following the five-component basic layer sequence.

**Figure 2 nanomaterials-10-00853-f002:**
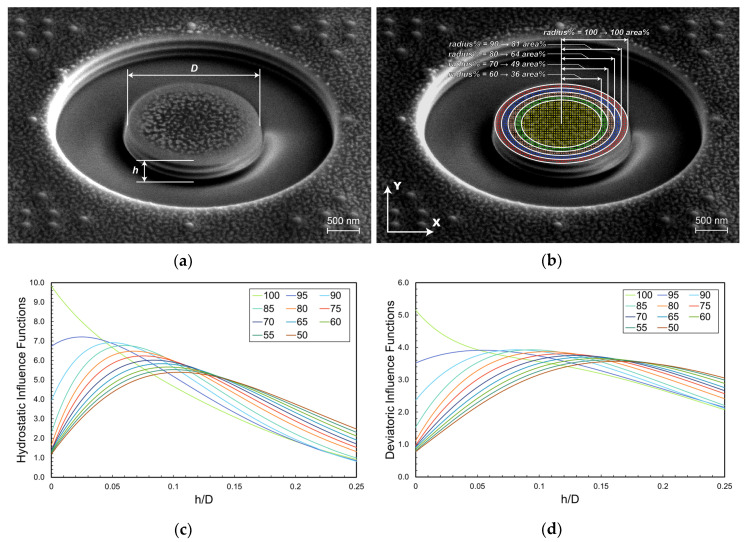
(**a**) Experimental example of ring-core focused ion beam (FIB) milling, (**b**) the idea of multiple *area%* digital image correlation (DIC) analysis for improving depth resolution and surface sensitivity, (**c**) FEM calculated hydrostatic part of the kernel function ***F(z/D)*** for several values of the *radius%*, (**d**) deviatoric components of the kernel functions for several values of the *radius%* (which defines the corresponding *area*% for DIC analysis, as reported in Figure 2a,b).

**Figure 3 nanomaterials-10-00853-f003:**
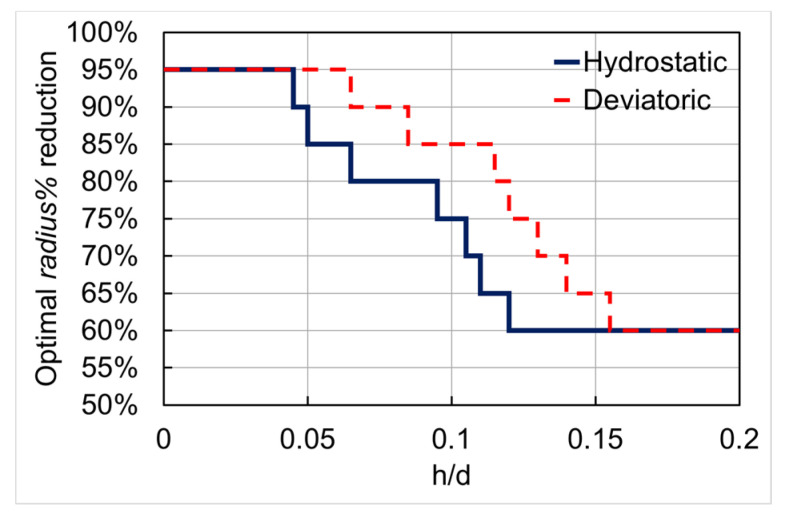
Optimal *radius%* (corresponding to *area%* for DIC) as a function of relative milling depth ***h*/*D*** (dashed line: deviatoric part, full line: hydrostatic part).

**Figure 4 nanomaterials-10-00853-f004:**
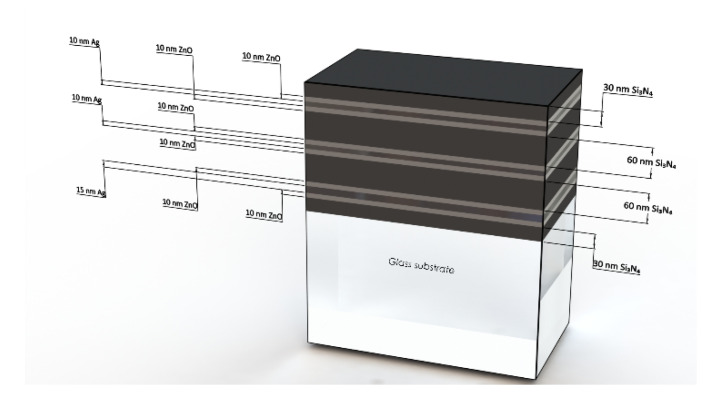
Schematic representation of the adopted film architecture, which is also visible in Figure 2a.

**Figure 5 nanomaterials-10-00853-f005:**
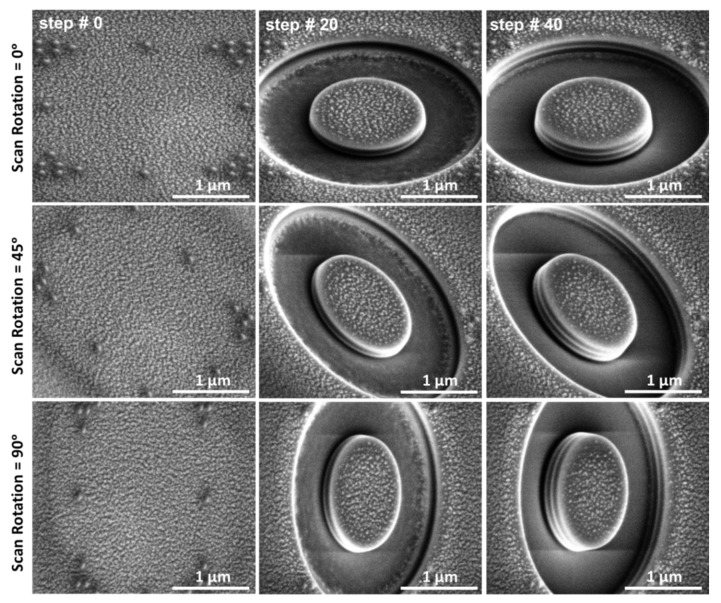
Step-by-step FIB milling of a 1.5 µm pillar on the coating. Surface features are gold patterning to facilitate the DIC procedure. The different layers are visible in the micrographs.

**Figure 6 nanomaterials-10-00853-f006:**
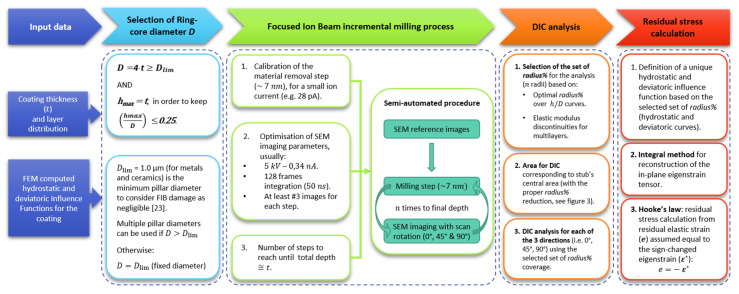
Flowchart describing the entire workflow of the method, with main decision steps on how to select the core diameter, FIB/SEM parameters, DIC analysis, and final residual stress calculation.

**Figure 7 nanomaterials-10-00853-f007:**
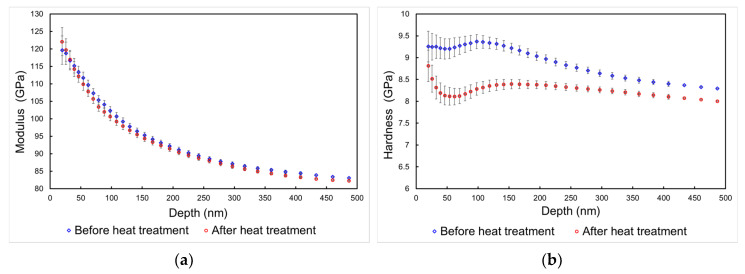
Nanomechanical testing on the film before and after the heat treatment. (**a**) Nanoindentation modulus profile, (**b**) nano-hardness profile, (**c**) SEM micrograph of a nanoscratch test BEFORE the heat treatment, (**d**) SEM micrograph of a nanoscratch test AFTER the heat treatment.

**Figure 9 nanomaterials-10-00853-f009:**
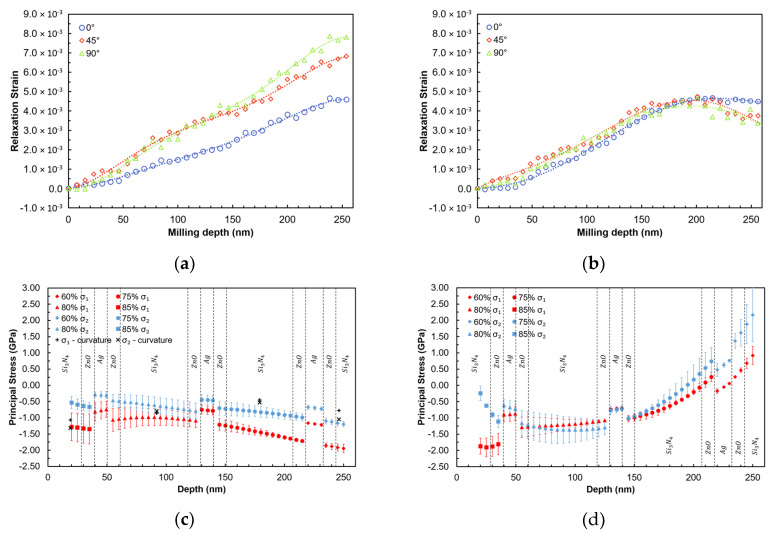
Analysis of the surface residual stress gradient for the sample after heat treatment. (**a**) Experimental relaxation strain (along one direction, as an example, where dashed lines represent the polynomial interpolation) for different values of *radius%*; (**b**) Corresponding residual stress profiles (second principal stress, as an example).

**Figure 8 nanomaterials-10-00853-f008:**
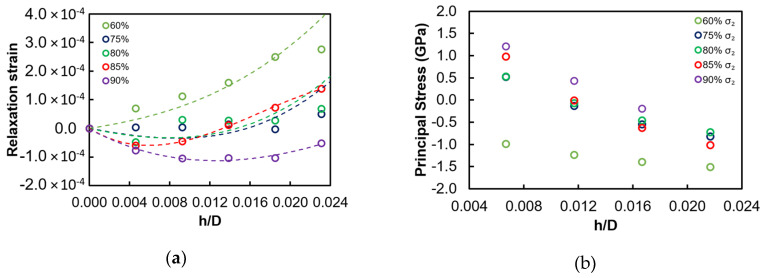
Relaxation strain (with fitting of polynomial function, grade 4) along three directions before heat treatment (examples for *radius%* = 80%) (**a**) and after heat treatment (**b**). Principal residual stress profiles (σ1 and σ2) obtained by considering multiple *radius%*, (**c**) before the heat treatment cycle and (**d**) after the heat treatment cycle. (Dashed lines indicate the different layers of the films, according to the architecture described in Section 2.2).

**Table 1 nanomaterials-10-00853-t001:** Result summary on nanoindentation and nanoscratch experiments.

Sample	Critical Load Lc2 (First Delamination, mN)	Critical Load Lc3 (Full Coating Delamination, mN)	Average Hardness (GPa) (Depth Range 25–35 nm)	Average Elastic Modulus (GPa) (Depth Range 25–35 nm)
BEFORE heat treatment	4.39 ± 0.36	17.66 ± 0.76	9.2 ± 0.4	117 ± 3
AFTER heat treatment	2.25 ± 0.13	15.80 ± 0.09	8.4 ± 0.4	118 ± 5

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
