# Peer review of "Nano-Scale Residual Stress Profiling in Thin Multilayer Films with Non-Equibiaxial Stress State"

_nanomaterials, 2020, doi:10.3390/nano10050853_

Round 1
Reviewer 1 Report
This paper presents a novel method to calculate residual stress using ring-shape trenches and a novel analysis using different areas of the images for DIC. I believe that the method is certainly interesting, and it means an improvement of the previous approaches. However, the description of the method is still fuzzy, and the conclusions not too solid. Therefore, I would recommend the following major revision.
- My main criticism, which also is behind my overall evaluation, is that the method is not sufficiently described in order to serve any other readers to apply this method. For instance, in too many places the paper references to previous works. In contrast, a paper of this nature MUST include all the elements to facilitate the application and reproduction of the methods by any other researcher without going to any other source. This is not negotiable. The authors do not need to add the full content of other papers, that’s clear, but all the equations, assumptions, etc. need to be included in the paper. Furthermore, it is not clear to the reader the ‘path’ to apply this method (see other comments). The authors must provide a ‘map’ or diagram that allows a user with a film of a certain thickness to know what to do, i.e. which dimension of the pillar, depth of analysis, function to use, number of images and erosions, etc. Furthermore, the paper presents results of 2 samples before and after heating, and only 1 ring of a certain size. I would strongly appreciate the comparison with rings of other diameters (i.e. check the consistency of the method), and also linear trenches (in both directions). The comparison with the curvature method, although interesting, it is not enough in my opinion, and in fact it does not validate one of the main observations (non-equibiaxial stress state).
- The abstract should be more concise and summarize the objective of the work. The real abstract starts in line 28, where the concept of the paper is presented. Please, re-organize the abstract to reduce its length and present clearly the novelty of the paper and the improvements towards previous cases.
- Lines 14-21 must be reduced to 2 lines max, and leave the rest to the introduction.
- Line 29: “The films were deposited by magnetron sputtering onto glass substrate and post deposition annealed at 540 °C for 12 minutes”. This information is irrelevant in an abstract.
- “We propose an innovative approach to improve depth resolution and surface sensitivity for residual stress depth profiling of thin Si3N4/Ag/ZnO films.” Please, provide the critical details about the innovations of the method.
- Lines 45-53: please add an scheme of a ‘typical’ staking of layers, and approximate thicknesses of each layer. Maybe re-location of Fig.3 could work, although maybe a more generic version would be better.
- Lines 53-63: please provide some values of the mechanical properties (just to give an idea to the readers), and also about the ‘high temperatures’ used in tempering.
- Line 66: as far as I know, the wafer curvature method can be used in films not amorphous. Please, correct.
- All the terms in Eq. 1 MUST be defied, in particular, z, D and h.
- Please, indicate how the ‘optimal % of area’ in Fig. 2 is selected. Is it that where the higher value in Figs. 1 c and d are obtained?
- Although I guess that I understand the idea of the method, I strongly recommend a Figure (flow diagram?) that would explain clearly the ‘sequence of events’ for e.g. a case of 300 nm. Then, you would need to define the diameter of the pillar(s), and afterwards the depth of interest would be achieved by proper selection of the DIC area. Am I right? In any case, please create a figure that explains this process properly.
- “In the case of multilayer films, the transition from one area% value to the next one can be set to correspond to the discontinuity of material property, namely, Young’s modulus”. This sentence is confusing. Please, clarify.
- “High resolution SEM micrographs were acquired before and after every milling step using an integral of 128 images and at 50 ns dwell time while maintaining the same contrast as the reference 230 image.”. Please, clarify the operation voltage of the SEM, and also the meaning of ‘every milling step’. Clarify what is the maximum depth achieved, and how many images were taken in the process. Further, I would like to know how do the authors know precisely the depth eroded in each step using SEM.
- Which DIC software has been used, and which parameters employed?
- “suitable elastic modulus and Poisson ratio values of individual layers were taken from literature [7,8].” Please, indicate those values, and also the meaning of ‘suitable’ in this context.
- “With the assumption that the residual stress state of the inner layers does not change after the deposition of additional layers”. This assumption seems pretty strong. Please, discuss a little about its accuracy.
- Figure 6: reduce the number of significant digits in the x and y axis, and increase the size of the numbers. Make sure that ticks appearing each 10 nm are included in the X axis, and make sure that these ticks also appear in the bottom of the Figure. Please, also label the different layers of the films (Si3N4, Ag and ZnO).
- Lines 328-332: it is indicated that the comparison with the curvature method is presented against different depths. However, I have several doubts about that; first, I do not understand why the curvature method does not distinguish between both directions, while the strain measurement finds a clear difference in the sample before heating. How can this be? Second, I do not see the point of comparison between both measurements of stress; I assume that the curvature method was taken from films grown with different thicknesses (which it is not explained in the text, by the way, and it should be). However, I thought that the values of stress depending on the depth were stress at ‘that depth’, i.e. not the stress that a film of a certain thickness would have. In other words, I understood that the stress at 200 nm depth is the stress value at that point, not the stress value of a film of 100 nm (assuming a total thickness of 300 nm). If the stress values given by the method are punctual, then I cannot understand the comparison with the values from the curvature method, unless they are ‘summed’ and compared with the curvature values. In any case, all of this has to be better explained in the text.
- Please, explain why there are no curvature measurements for the film after the heat treatment.
- The ‘jumps’ in Fig. 6 connected to the Ag layers are only due to the value of Elastic modulus used, right? I assume that the value of Ag used is pretty different from Si3N4 and ZnO. Please, state that in the paper and include the values.
- “(i) coating adhesion depends strongly on the contact radius of the counterpart during wear”. What? The adhesion of the coating to the substrate should be a certain value, regardless of the way of measurement. Please, explain.
- Lines 336-338: These lines are confusing; please explain.
- In connection with the 2 previous points, I believe that the authors confuse ‘adhesion’ with ‘critical load’. The first concept is ‘what it is’. The second refers to a parameter that it is directly connected to the measurement, and I can certainly believe that a film with a certain adhesion may show different critical loads depending on the radius, and also that the radius of the tip can change the depth where is a maximum stress under the same load. However, that does not mean that the adhesion of the film is different.
- 7: explain the meaning of the dashed lines in the caption. Use the same symbols in Fig. 7a than in Fig. 7b. Add the relaxation strains for the 90%, since the values of stress appear in Fig. 7b.
- “In particular, using 85% could capture the surface stress gradient for the sample after heat treatment, whilst the use of lower area% values does not capture the tensile stress at film surface”. That sentence is difficult to understand to me; if the ‘truth’ are the values obtained at 80% of area, then the values of stress for the area 85% are very different in the surface (0.5 GPa at 80% vs. 1 GPa at 85%, the double). In contrast, it seems that the values obtained from 75% are pretty close for all depths. Please, explain.
- What is the fundamental reason why 80% area gives ‘the true values’? I expected lower error bars for that area selection since it offers the best sensitivity in Eq. 1, but I did not expect such big changes. Please, clarify.
- Lines 400-404: then what is the alternative? Make many rings in different locations of the sample?
Author Response
1a. My main criticism, which also is behind my overall evaluation, is that the method is not sufficiently described in order to serve any other readers to apply this method. For instance, in too many places the paper references to previous works. In contrast, a paper of this nature MUST include all the elements to facilitate the application and reproduction of the methods by any other researcher without going to any other source. This is not negotiable. The authors do not need to add the full content of other papers, that’s clear, but all the equations, assumptions, etc. need to be included in the paper.
The authors thank the reviewer for this very reasonable comment. We have revised the entire paper and added details on all experimental methodologies and calculation procedures, in order to make it self-consistent and completely independent from previous articles. Clearly, the references to previous articles remain in order to make it clear on what are the new parts with respect to our previous works. All revisions are highlighted in yellow in the paper.
1b. Furthermore, it is not clear to the reader the ‘path’ to apply this method (see other comments). The authors must provide a ‘map’ or diagram that allows a user with a film of a certain thickness to know what to do, i.e. which dimension of the pillar, depth of analysis, function to use, number of images and erosions, etc.
Thank you for the useful comment. We have added a scheme reporting the characteristic workflow of the method, describing all steps required for selection of the ring diameter. Everything is normalised with respect to film thickness t. We would also like to refer the Reviewer (and readers) to the Good Practice Guide that details the practical steps to the application of this method published by the National Physical Laboratory (NPL), UK[1].
1c. Furthermore, the paper presents results of 2 samples before and after heating, and only 1 ring of a certain size. I would strongly appreciate the comparison with rings of other diameters (i.e. check the consistency of the method), and also linear trenches (in both directions).
The case reported in this paper is focused on thin films with thickness < 300 nm. In this case, the possibility of using pillars with multiple diameters (as explored in detail in previous papers from the authors[2]) was not possible, given the likely artefacts coming from FIB milling of pillars with diameter < 1 µm (a practical lower limit to the ring-core diameter). That is exactly the reason why we arrived to the idea of using multiple values of area%, which lead to very good agreement of the stress data obtained from the different %area considered and allowed to improve the surface sensitivity and the maximum achievable depth for the method.
A review paper from some of the authors[3] considered different trench shapes and came to the conclusion that circular geometry offers significant advantages for residual stress depth profiling. In fact, linear trenches are unlikely to deliver the sensitivity sought in this study, as relief strain is highly non-uniform and the mathematical framework to extract the depth variation of eigenstrain / residual stress is not available.
Therefore, the authors are convinced that adopting different geometries would not strengthen method validation. A discussion of this point has been added in the paper.
1d. The comparison with the curvature method, although interesting, it is not enough in my opinion, and in fact it does not validate one of the main observations (non-equibiaxial stress state).
Validation of the obtained results is a very critical issue, given the complexity of the system under evaluation (thin amorphous/crystalline multilayer films over an amorphous substrate). The improved curvature method that we have adopted was the only method found suitable to obtain (at least) some average values for the residual stress at different depths for the film. As reported in §2.5, the curvature method was extended in order to make an analysis on non-biaxial stress states at different depths.
The agreement that we found between the two methods is very good and falls inside the statistical uncertainty of the two methods. In addition, it is not easy to carry out curvature measurements in different directions, in order to get more information on non-biaxial stress, because squared samples would have anticlastic effects. Therefore, we believe the provided validation is the best we could do with respect to the extreme complexity of the samples under investigation.
- The abstract should be more concise and summarize the objective of the work. The real abstract starts in line 28, where the concept of the paper is presented. Please, re-organize the abstract to reduce its length and present clearly the novelty of the paper and the improvements towards previous cases.
- Lines 14-21 must be reduced to 2 lines max, and leave the rest to the introduction.
- Line 29: “The films were deposited by magnetron sputtering onto glass substrate and post deposition annealed at 540 °C for 12 minutes”. This information is irrelevant in an abstract.
- “We propose an innovative approach to improve depth resolution and surface sensitivity for residual stress depth profiling of thin Si3N4/Ag/ZnO films.” Please, provide the critical details about the innovations of the method.
The authors thank the reviewer for the recommendations on the abstract contents and incisiveness and have accordingly re-organized its structure and improved its conciseness, providing also the crucial points of the developed approach, with the major modifications been highlighted. Coating details have been also removed and left to be addressed in the introductory section of the article, for which modifications have been highlighted in yellow.
- Lines 45-53: please add a scheme of a ‘typical’ staking of layers, and approximate thicknesses of each layer. Maybe re-location of Fig.3 could work, although maybe a more generic version would be better.
The authors thank the reviewer for the recommendation. A scheme for the typical staking sequence of dielectric (Si3N4 and ZnO in the specific case of the coatings studied in the present work) and reflective layers (Ag) have been inserted and referenced in the text. That is a five-component stack is which dielectric layers are placed at the bottom and top of the reflective one, providing optical filling anti-reflective functions, adhesion and protection. The typical value of thickness adopted for the reflective layer have been also provided, as recommended by the reviewer, while thicknesses of the dielectric ones depend upon the overall coating architecture (basing on target performance) and are specifically reported in the scheme of the coatings studied in this work. Furthermore, the sequence represents the basic building block for coatings with multiple reflective layers.
- Lines 53-63: please provide some values of the mechanical properties (just to give an idea to the readers), and also about the ‘high temperatures’ used in tempering.
The authors thank the reviewer for the useful recommendation. Details of the tempering process typical temperatures have been provided along with maximum set point values for the furnace. As the mechanical properties for those coatings strictly depend upon the dielectric layer materials, their stack sequence, overall coating architecture and sputtering parameters, references to works presenting significative data are provided. Specific indications for the coatings studied in the present work have been provided in the Materials & Methods section as suggested also by point 12 recommendation from the reviewer.
- Line 66: as far as I know, the wafer curvature method can be used in films not amorphous. Please, correct.
Actually, the curvature method (Stoney equation) is perfectly suitable for amorphous thin films. The only problems related to residual stress measurement by curvature can arise for the cases where the substrate (or the film) are characterised by a certain level of elastic anisotropy (e.g. in case of single crystal). Such a situation (NOT present in this study) can be corrected by using modified forms of the Stoney’s equation.
- All the terms in Eq. 1 MUST be defied, in particular, z, D and h.
The authors thank the reviewer for highlighting the missing definitions of terms in equation 1, that have been accordingly inserted in the highlighted text.
- Please, indicate how the ‘optimal % of area’ in Fig. 2 is selected. Is it that where the higher value in Figs. 1 c and d are obtained?
Yes, the optimal area% curves in what now is Figure 3 are computed basing on the range of relative milling depth (h/D) for which the strain sensitivity functions values associated with an area% in Figure 2 c and d are higher than other. Seizing this question, the authors have inserted a sentence that clarifies this point also in the text.
- Although I guess that I understand the idea of the method, I strongly recommend a Figure (flow diagram?) that would explain clearly the ‘sequence of events’ for e.g. a case of 300 nm. Then, you would need to define the diameter of the pillar(s), and afterwards the depth of interest would be achieved by proper selection of the DIC area. Am I right? In any case, please create a figure that explains this process properly.
The authors thank the reviewer for the extremely valuable suggestion and a diagram with complete workflow and guidelines has been added in the paper.
- “In the case of multilayer films, the transition from one area% value to the next one can be set to correspond to the discontinuity of material property, namely, Young’s modulus”. This sentence is confusing. Please, clarify.
The sentence has been modified accordingly to the recommendation. The idea behind the original text has been developed in detail in the highlighted reviewed paragraph: mathematical calculation discontinuities could occur for residual stress calculations if the h/D values, corresponding to changes of the optimal area% graph, fall within the h/D range of existence of a layer due to a change in the definition of the strain sensitivity function. This issue can be avoided by setting the area% transitions to correspond to Young’s modulus discontinuities at each interface from one layer to another, moderately offsetting from the optimal area% curves, still maintaining the improved strain sensitivity of the multiple-areas technique.
- “High resolution SEM micrographs were acquired before and after every milling step using an integral of 128 images and at 50 ns dwell time while maintaining the same contrast as the reference 230 image.”. Please, clarify the operation voltage of the SEM, and also the meaning of ‘every milling step’. Clarify what is the maximum depth achieved, and how many images were taken in the process. Further, I would like to know how do the authors know precisely the depth eroded in each step using SEM.
The authors have included in the paper additional information on the image acquisition parameters for the SEM, as recommended by the reviewer, as well as clarifications on the notion of “milling step”. In particular, the “incremental material removal” concept has been introduced in the text to elucidate the term “step” as the partial increment contributing to the total milling depth reached. Five images have been acquired of the original surface and of the surface after each partial milling (step) to track strains on the pillar surface with respect to the reference. Calibrations of the milling depth vs ion dose have been performed on the tested materials using the same milling geometries adopted for the full tests and performing measurements on cross-sections over the calibration pillars. Using this approach and extremely reduced ion currents ensures, for the geometries under investigation, extremely small incremental milling steps resulting in uniform and constant incremental depths. The final depth reached, that is about the coating thickness to evaluate the residual stress depth profile from the surface to the interface with the substrate, is therefore derived from the total number of steps. Cross-checking via cross-sectional depth measurements on SEM high resolution imaged of the milled geometries lateral surfaces have been also performed. Further explanation has been inserted in the highlighted text to provide clarifications as recommended by the reviewer.
- Which DIC software has been used, and which parameters employed?
An open source MATLAB™ based DIC routine have been used, for which the authors have inserted proper reference in the paper. The main parameters used during the image correlation analysis have been also added to the text.
- “suitable elastic modulus and Poisson ratio values of individual layers were taken from literature [7,8].” Please, indicate those values, and also the meaning of ‘suitable’ in this context.
Thank you for this relevant remark. The authors have reported in the text the values of the elastic modulus and Poisson’s ratio used for the residual stress measurement campaign. Nanoindentations were performed on the multilayer coating systems using the Continuous Stiffness Measurement (CSM) mode. Deconvolution models have been applied to the composite modulus data to evaluate the value of this property for each layer of the film, specifically Dorner-Nix, Korsunsky and Puchi-Cabrera models for multilayer films. Given the complex architecture of the film, particularly due to the extremely challenging 10 nm thickness of the reflective layers, it was not possible to have a statistically robust evaluation and, therefore, values has to be validated trough literature review for thin films of the same kind as reported in the references: hence the term “suitable” in the text.
- “With the assumption that the residual stress state of the inner layers does not change after the deposition of additional layers”. This assumption seems pretty strong. Please, discuss a little about its accuracy.
The author thank the reviewer for the valuable discussion point, this is clearly a limitation of the curvature method for measuring residual stress gradients in coatings. The limitation of conventional methods is (actually) the main motivation and the main challenge at the basis of the current research. We added a comment in the paper.
- Figure 6: reduce the number of significant digits in the x and y axis, and increase the size of the numbers. Make sure that ticks appearing each 10 nm are included in the X axis, and make sure that these ticks also appear in the bottom of the Figure. Please, also label the different layers of the films (Si3N4, Ag and ZnO).
Figure 6, now updated to Figure 8 following the reviewer recommendation points, have been modified accordingly.
- Lines 328-332: it is indicated that the comparison with the curvature method is presented against different depths. However, I have several doubts about that; first, I do not understand why the curvature method does not distinguish between both directions, while the strain measurement finds a clear difference in the sample before heating. How can this be? Second, I do not see the point of comparison between both measurements of stress; I assume that the curvature method was taken from films grown with different thicknesses (which it is not explained in the text, by the way, and it should be). However, I thought that the values of stress depending on the depth were stress at ‘that depth’, i.e. not the stress that a film of a certain thickness would have. In other words, I understood that the stress at 200 nm depth is the stress value at that point, not the stress value of a film of 100 nm (assuming a total thickness of 300 nm). If the stress values given by the method are punctual, then I cannot understand the comparison with the values from the curvature method, unless they are ‘summed’ and compared with the curvature values. In any case, all of this has to be better explained in the text.
This is a very good observation from the reviewer. As also described in previous points, the curvature method is based on the strong assumption that the residual stress state of the inner layers does not change after the deposition of additional layers. This can be a main explanation on why the curvature method is not able to capture the non-biaxiality of the stress. Additionally, the non-biaxiality could also arise during the final cooling of the large area (thick) substrates. This point is very relevant and has been added in the discussion section.
The curvature method is NOT simply considering the stress at a certain film thickness, but it can reconstruct the stress state AT a certain depth, by using the equations 6-7 in the paper. The only assumption (that we already discussed to be a limitation) is that the residual stress state of the inner layers does not change after the deposition of additional layers.
Therefore, the two methods are comparable, but the reviewer is right to ask a deeper discussion on this (which was added in the discussion section).
- Please, explain why there are no curvature measurements for the film after the heat treatment.
The curvature measurements are performed on thin-glass-substrate, which are not subjected to the heat treatment process. This is because the heat treatment is very close to the industrial treatment and is not suitable for the thin glass substrates.
- The ‘jumps’ in Fig. 6 connected to the Ag layers are only due to the value of Elastic modulus used, right? I assume that the value of Ag used is pretty different from Si3N4 and ZnO. Please, state that in the paper and include the values.
The authors thank the reviewer for the observation and to clarify this point in the paper values of elastic moduli of all layers are now reported, also following indications from previous discussion.
- “(i) coating adhesion depends strongly on the contact radius of the counterpart during wear”. What? The adhesion of the coating to the substrate should be a certain value, regardless of the way of measurement. Please, explain.
We thank the reviewer for this relevant comment. The authors refer to the "critical load from scratch” when talking about adhesion. We agree with the reviewer that the use of the word “adhesion” is not correct in the paper and we have replaced it with the term “critical load”.
- Lines 336-338: These lines are confusing; please explain.
The sentence refers to the correlation between residual stress depth gradient and coating adhesion measured by scratch testing. We have modified it in the paper in order to be clearer.
- In connection with the 2 previous points, I believe that the authors confuse ‘adhesion’ with ‘critical load’. The first concept is ‘what it is’. The second refers to a parameter that it is directly connected to the measurement, and I can certainly believe that a film with a certain adhesion may show different critical loads depending on the radius, and also that the radius of the tip can change the depth where is a maximum stress under the same load. However, that does not mean that the adhesion of the film is different.
The authors fully agree with the reviewer. We have corrected the text in order to use “critical load” when necessary.
- 7: explain the meaning of the dashed lines in the caption. Use the same symbols in Fig. 7a than in Fig. 7b. Add the relaxation strains for the 90%, since the values of stress appear in Fig. 7b.
Dashed lines in 7a (which now has become 9a) simply represent the polynomial interpolation of the relaxation strain data (we have added this relevant info in the figure caption).
Figures 9a and 9b are now fully consistent in terms of data and symbols.
- “In particular, using 85% could capture the surface stress gradient for the sample after heat treatment, whilst the use of lower area% values does not capture the tensile stress at film surface”. That sentence is difficult to understand to me; if the ‘truth’ are the values obtained at 80% of area, then the values of stress for the area 85% are very different in the surface (0.5 GPa at 80% vs. 1 GPa at 85%, the double). In contrast, it seems that the values obtained from 75% are pretty close for all depths. Please, explain.
We thank the reviewer for the valuable discussion point. As it will be explained also in point 23 from reviewer’s recommendations, the 80% area does not give the true value for residual stresses, but it represents the area suggested in previous papers as by so is a reference for this work. The important point is exactly what has been stated by the reviewer that 85% (higher values) are different in the surface and, actually, more reliable at the shallowest depths, while lower values (75%) enhance the range of explorable depths (range of profiling). This is also the reason why 75% data looks smoother than 85%: in fact, using 85% area allows for a better sensitivity to the near-surface stress gradients, which are not captured well by the 75% area. On the contrary, 75% allows to have reliable data at the deepest depths (h/D up to 0.25).
- What is the fundamental reason why 80% area gives ‘the true values’? I expected lower error bars for that area selection since it offers the best sensitivity in Eq. 1, but I did not expect such big changes. Please, clarify.
80% is not the true value, but just the value that was suggested in previous papers to give (in average) robust results. In this paper, we found that HIGHER values (e.g. 85%) are actually more reliable at the shallowest depths, while lower values (e.g. 75%) allow to improve the maximum explorable depth.
- Lines 400-404: then what is the alternative? Make many rings in different locations of the sample?
We have modified and clarified the sentence. (Actually, here we are referring to the comparison between the old classical method and the new one).
[1] https://www.npl.co.uk/special-pages/guides/gpg143
[2] A.M. Korsunsky, E.Salvati, A. J. Lunt, …, M. Sebastiani, Nanoscale residual stress depth profiling by Focused Ion Beam milling and eigenstrain analysis, Materials & Design, https://doi.org/10.1016/j.matdes.2018.02.044
[3] A.J.G. Lunt, A.M. Korsunsky, A review of micro-scale focused ion beam milling and digital image correlation analysis for residual stress evaluation and error estimation. Surface & Coatings Technology (2015) http://dx.doi.org/10.1016/j.surfcoat.2015.10.049

Reviewer 2 Report
Nano-scale residual stress profiling in thin multilayer films with non-equibiaxial stress state
The authors state that this is a unique method that is a combination of focused ion beam (FIB) milling and digital image correlation (DIC) for accurately probing non-equiaxial residual stresses with nanoscale resolution in thin films. Although I think this paper may be interesting to readers of Nanomaterials, there may be few things to be solved as shown below:
- The main idea is the use of different “area%” (Figs. 1 and 2) for DIC evaluation at different milling depths to maximize strain sensitivity. However, there may be a strong objection to this idea. Round side surfaces of pillars are free surfaces with no tractions. These free surfaces affect elastic deformation behavior on top surfaces of the pillars near the side surfaces. This means that the “area%” dependence may come from the effects of the free surfaces. I think this should be discussed in this paper.
Find below some other comments to be addressed.
- In scientific journals, it may be unusual that a paragraph is composed of a sentence. However, this occurs in some parts of this paper such as lines from121- 174 (page 4) and lines from 362 to 373 (page 11).
- The caption of Fig. 5: (c) of “.., (c) SEM ...” should be (d).
- The caption of Fig. 6: (A), (B), (C) and (D) should be (a), (b), (c) and (d).
Author Response
- The main idea is the use of different “area%” (Figs. 1 and 2) for DIC evaluation at different milling depths to maximize strain sensitivity. However, there may be a strong objection to this idea. Round side surfaces of pillars are free surfaces with no tractions. These free surfaces affect elastic deformation behavior on top surfaces of the pillars near the side surfaces. This means that the “area%” dependence may come from the effects of the free surfaces. I think this should be discussed in this paper.
We thank the reviewer for the useful comment. Yes, the effects on surface strain distribution over pillar’s surface have been fully accounted for in our analyses. As shown in one of our previous papers, gradients of deformation, and obviously gradients of stress as well, are present particularly close to the edge free surface, as the reviewer pointed out. Relaxation strain profiles at different milling depths are shown in this Figure[1]:
This is exactly where the “area%” dependence comes from, as suggested by the reviewer. One of the novelties of this paper is indeed the exploitation of this characteristic to improve the surface sensitivity and maximise the maximum explorable depth, by using different area% values at different depths.
It is worth mentioning that we are also aware of the possible damage induced by the FIB milling at the edges of the pillar. In fact, the area% values employed is never higher than 90%.
We added some more details in the paper on these points.
- In scientific journals, it may be unusual that a paragraph is composed of a sentence. However, this occurs in some parts of this paper such as lines from121- 174 (page 4) and lines from 362 to 373 (page 11).
Issues related to the structure of the paragraphs have been fixed as suggested by the reviewer.
- The caption of Fig. 5: (c) of “.., (c) SEM ...” should be (d).
The authors have fixed this issue in the figure, now updated to Figure 6.
- The caption of Fig. 6: (A), (B), (C) and (D) should be (a), (b), (c) and (d).
The authors thank the reviewer for highlighting the issue that has been fixed accordingly.
[1]A.M.Korsunsky et al. Nanoscale residual stress depth profiling by Focused Ion Beam milling and eigenstrain analysis. JMAD 2018.

Round 2
Reviewer 1 Report
The authors have provided convincing responses to my previous review and they have also improved the quality of the paper a lot, since its readability has improved clearly. Therefore, I would recommend this minor revision:
Abstract: the line 30-36 is composed by one sentence. Please, divided it. Furthermore, this sentence is presented, it seems that this method only serves for that multilayer, while I understand that it serves for many cases and it is illustrated using a particular stacking as an example. Please, re-write.
Figure 6: this Figure is excellent, and very illustrative of the overall process that is needed to carry out this method. I do not understand few points, though:
- In the top square in the ‘Selection of the ring core diameter’. Why t/0.25 instead 4*t?. ‘where (h/D)=0.25’ does not apply here, since in the previous expression there is no (h/D). Finally, if h/D=0.25 and D=t/0.25, why not saying that hmax=t (which is obvious by the way)? What do I miss here?
- In the FIB process, the boxes are numbered from top to bottom as 1, 3 2. Is that a mistake?
- I guess that the authors should refer to Fig. 3 in the top box of ‘DIC analysis’.
Response to my previous point #5: I know that curvature method applies to amorphous films. However, considering how it is written lines 85-86 (in the present version of the paper), it seems to me that the curvature method does only work for amorphous films. This is just a very minor point, leave the text as it is if you prefer so.
Author Response
The authors thank the reviewer for the suggestions, which have been performed and highlighted in yellow in the paper, namely:
- Line 30-36 of the abstract are now composed of three sentences, and the text has been modified in order to show that the method is suitable for a general thin film, including multilayers.
- Figure 6 has been revised according to suggestions.
Reviewer 2 Report
The revision is satisfactory and I recommend the publication of this papaer in Nanomaterials.
Author Response
The authors are very thankful to the reviewer for the positive evaluation.